# Organocatalytic enantio- and diastereoselective cycloetherification via dynamic kinetic resolution of chiral cyanohydrins

Naoki Yoneda[1], Yuki Fujii[1], Akira Matsumoto[1], Keisuke Asano [1] & Seijiro Matsubara [1]

Enantioselective approaches to synthesize six-membered oxacycles with multiple stereogenic centres are in high demand to enable the discovery of new therapeutic agents. Here we present a concise organocatalytic cycloetherification for the highly enantio- and diastereoselective synthesis of tetrahydropyrans involving simultaneous construction of two chiral centres, one of which is fully substituted. This method involves dynamic kinetic resolution of reversibly generated chiral cyanohydrins. A chiral bifunctional organocatalyst selectively recognizes a specific chair-like conformation of the intermediate, in which the small steric effect of the linear cyano group as well as its anomeric effect play important roles in controlling stereoselectivity. The products offer additional utility as synthetic intermediates because the cyano group can be further transformed into a variety of important functional groups. This strategy provides a platform to design efficient approaches to obtain a wide range of optically active tetrahydropyrans, which are otherwise synthetically challenging materials.

[1] Department of Material Chemistry, Graduate School of Engineering, Kyoto University, Kyotodaigaku-Katsura, Nishikyo, Kyoto 615-8510, Japan. Correspondence and requests for materials should be addressed to K.A. (email: asano.keisuke.5w@kyoto-u.ac.jp) or to S.M. (email: matsubara.seijiro.2e@kyoto-u.ac.jp)

The relative stereochemistry of saturated six-membered cyclic compounds has become one of the most established concepts in the conformational analysis of organic molecules since the pioneering work of Barton[1] and Hassel[2]. The saturated rings generally adopt stable chair conformations of unstrained sp[3] hybrid atoms, with bulky substituents preferring to reside in equatorial positions to minimize steric clashes. Effective orbital interactions can also stereoelectronically control the relative configurations. Additionally, the absolute configurations of these frameworks are also a good opportunity for interaction with chiral hosts. In fact, optically active tetrahydropyrans (saturated six-membered oxacycles) are ubiquitous scaffolds in a wide range of bioactive compounds[3–6], and their biological activities are strongly affected by their absolute stereochemistry. Thus, optically active derivatives are expected to have unexploited pharmaceutical activities; however, the lack of a simple robust method for their enantioselective synthesis has limited their development. In particular, as the enantio- and diastereoselective construction of multiple stereocentres in a single operation often poses a formidable challenge, it remains desirable to develop a concise, efficient method for the asymmetric installation of more than one chiral centre in tetrahydropyrans.

Cycloetherification of secondary or tertiary alcohols bearing an unsaturated moiety affords oxacycles containing two stereogenic centres via kinetic resolution of the racemic alcohols[7–9] (Fig. 1a). Cyclization via dynamic kinetic resolution involving epimerization of chiral alcohols, enabling quantitative yields of the desired product, is more desirable; however, it has not yet been achieved. Epimerization of tertiary alcohols, which cannot be oxidized, requires especially harsh reaction conditions that are not suitable for asymmetric catalysis. To realize the cyclization of chiral tertiary alcohols via dynamic kinetic resolution, we propose a process involving reversible addition of a carbon nucleophile to ketones followed by cyclization[10], leading to the efficient simultaneous construction of two stereogenic centres, including a tetrasubstituted chiral carbon (Fig. 1b). Furthermore, to accomplish the stereoselective construction of a tetrasubstituted chiral centre, which has been a long-standing challenge in organic synthesis[11–13], we aimed to use a small electronegative carbon nucleophile. These features favour the introduction of the substituent adjacent to the heteroatom in an axial position in a six-membered oxacycle, enabling weaker 1,3-diaxial interactions as well as a favourable orbital interaction with the oxygen atom (anomeric effect)[14].

## Results

**Reaction design.** Based on the previously mentioned concepts, we selected hydrogen cyanide as a suitable carbon nucleophile because the stereoselective cyanation of ketones is an efficient method to construct tetrasubstituted chiral centres[15–31], and the cyano group is known to have a small A value (conformational energy)[32] and is capable of inducing an anomeric effect because of its electronegativity[33–35]. The diverse chemistry of the cyano group also expands the utility of the resulting products as synthetic intermediates[36, 37]. Thus, the proposed dynamic kinetic resolution of chiral cyanohydrin intermediates, which are generated reversibly in situ, accompanied by an asymmetric intramolecular oxy-Michael addition[38–48] mediated by bifunctional organocatalysts[49–53], should enable a concise enantioselective synthesis of tetrahydropyrans with two chiral centres, including one fully substituted stereogenic centre (Fig. 2a). A chiral bifunctional organocatalyst can hydrogen bond to a specific conformation from the isomers generated during interconversions between both enantiomers of the intermediates in various conformations[54]. This complex immediately catalyses the subsequent asymmetric oxy-Michael addition from the recognized chair-like conformation with the cyano group in the axial position (**A**), also favoured by the weak 1,3-diaxial interactions and anomeric effect, to simultaneously generate two stereogenic centres (Fig. 2b). The resulting cyclic structures are found in a variety of bioactive agents[3–6]; the functionality of the cyano group has not only been used for further transformations in the synthesis of important compounds[55–60], but also plays a major role in their biological activities[61–63].

**Optimization of reaction conditions.** We initially investigated a model system consisting of bis-ketone **1a**, acetone cyanohydrin (**2**) and 10 mol % of cyclohexanediamine-based aminothiourea catalyst **4a** in CH$_2$Cl$_2$ at 25 °C. As expected, the tetrahydropyran product was obtained quantitatively with excellent enantio- and diastereoselectivity (Table 1, entry 1). Catalyst **4a**, bearing a piperidyl group, was shown to be more effective to obtain higher reactivity and stereoselectivity than catalyst **4b** (Table 1, entries 1

**Fig. 1** Simultaneous construction of two stereogenic centres in tetrahydropyrans. **a** Cycloetherification via kinetic resolution of racemic alcohols. **b** Cycloetherification via dynamic kinetic resolution involving reversible addition of a carbon nucleophile to ketones

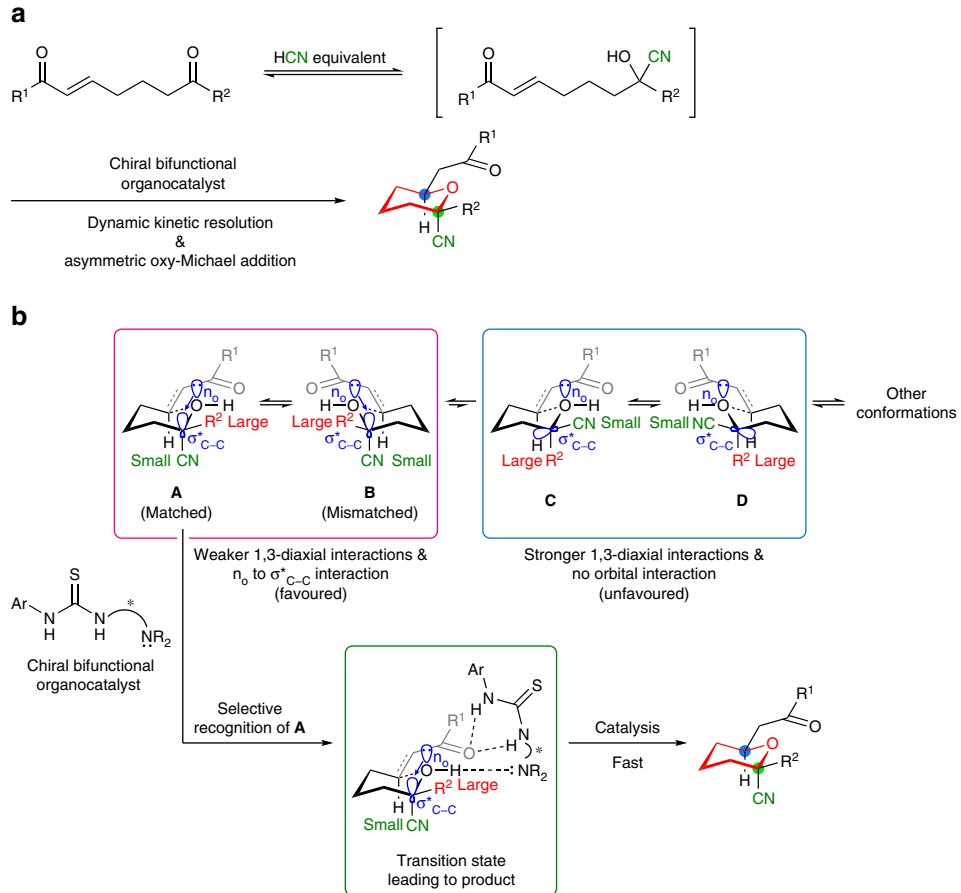

**Fig. 2** Reaction design for cycloetherification via dynamic kinetic resolution. **a** Intramolecular oxy-Michael addition via dynamic kinetic resolution through reversible generation of chiral cyanohydrins. **b** Rationale for the proposed strategy

and 2). Catalyst **4c**, which has a significantly less basic nitrogen atom, was not active, implying that the bifunctionality of catalysts containing amino and thiourea groups is important in this transformation (see also Supplementary Fig. 1–3). Cinchona alkaloid-derived aminothiourea catalysts were also shown to be effective, albeit with slightly lower reactivities and stereoselectivities (Table 1, entries 4–7). Alternative cyanide sources were also investigated. Trimethylsilyl cyanide in the presence of 2-propanol, which is known to generate hydrogen cyanide in situ[29], gave the same stereoselectivity but a slightly lower yield (Table 1, entry 8). The use of trimethylsilyl cyanide alone afforded the product with the same stereoselectivity but a much lower yield (Table 1, entry 9). Solvent optimization studies identified $CH_2Cl_2$, $CHCl_3$, and hydrocarbon solvents as affording especially high stereoselectivities with good yields (Table 1, entries 1, 10–12), while the use of polar solvents, which generally decrease anomeric effects, gave decreased yields and diastereoselectivities albeit with high enantioselectivities observed for both diastereomers (Table 1, entries 13–17). The reaction could also be carried out using a smaller amount of **2** and lower catalyst loading of **4a**, giving the same excellent stereoselectivity with a slight decrease of yield (Table 1, entry 18).

**Substrate scope.** With the optimized conditions (10 mol % catalyst **4a** in $CH_2Cl_2$ at 25 °C) in hand, we then explored the substrate scope (Fig. 3; see also Supplementary Fig. 4–6). Both electron-rich and -poor enones were tolerated, affording the corresponding products in high yields with good stereoselectivities (Fig. 3, **3b** and **3c**; see also Supplementary Fig. 7). An

enone bearing a heterocyclic ring gave comparable results (Fig. 3, **3d**), and an aliphatic enone provided the product in moderate yield with good stereoselectivity (Fig. 3, **3e**). Furthermore, an α,β-unsaturated thioester, which is useful for further transformations[40], afforded the product in good yield with high stereoselectivity (Fig. 3, **3f**). We went on to investigate the substituents that could be tolerated on the ketone (varying R'). A range of electron-rich and -poor aryl and heteroaryl ketones was tolerated, giving moderate to good yields, high enantioselectivities, and excellent diastereoselectivities (Fig. 3, 3g–3j). Aliphatic ketones were also successfully transformed, affording the desired products with high enantioselectivities and maintaining excellent diastereoselectivities (Fig. 3, **3k–3n**). It is noteworthy that the methyl-substituted ketone yielded the corresponding tetrahydropyran with high enantio- and diastereoselectivities despite the relatively small difference in size between methyl and cyano groups, reinforcing that assistance was provided by the anomeric effect (Fig. 3, **3n**; see also Supplementary Fig. 8).

**Mechanistic insights.** The absolute configuration of **3a** was determined by X-ray analysis (Fig. 3, see Supplementary Fig. 122 for details), and the configurations of all other materials were assigned analogously. As expected, the cyano group is located in the axial position of the chair-like oxacycle, consistent with our rationale for the reaction design (Fig. 2b). Additionally, to examine whether cyanohydrin formation proceeded enantioselectively in the presence of a chiral bifunctional catalyst[27, 28], we carried out a reaction using ketone **5**, which lacks an α,β-unsaturated carbonyl moiety, under the optimized conditions (Fig. 4).

**Table 1 Optimization of reaction conditions**

| Entry | Catalyst | Solvent | Yield (%) | dr | ee (%) |
|-------|----------|---------|-----------|-----|--------|
| 1 | **4a** | $CH_2Cl_2$ | 99 | > 20:1 | 97 |
| 2 | **4b** | $CH_2Cl_2$ | 95 | 14:1 | –97 |
| 3 | **4c** | $CH_2Cl_2$ | < 1 | — | — |
| 4 | **4d** | $CH_2Cl_2$ | 89 | 14:1 | –92 |
| 5 | **4e** | $CH_2Cl_2$ | 69 | 17:1 | –94 |
| 6 | **4f** | $CH_2Cl_2$ | 72 | 11:1 | 93 |
| 7 | **4g** | $CH_2Cl_2$ | 82 | 10:1 | 94 |
| 8[a] | **4a** | $CH_2Cl_2$ | 84 | > 20:1 | 97 |
| 9[b] | **4a** | $CH_2Cl_2$ | 14 | > 20:1 | 97 |
| 10 | **4a** | $CHCl_3$ | 93 | > 20:1 | 97 |
| 11 | **4a** | Benzene | 93 | > 20:1 | 95 |
| 12 | **4a** | Toluene | 90 | > 20:1 | 95 |
| 13 | **4a** | $Et_2O$ | 59 | 20:1 | 93 |
| 14 | **4a** | THF | 15 | 20:1 | 96 |
| 15 | **4a** | EtOAc | 38 | 17:1 | 94 |
| 16 | **4a** | $CH_3CN$ | 54 | 3.6:1 | 95 (93[d]) |
| 17 | **4a** | EtOH | 23 | 9.2:1 | 96 (88[d]) |
| 18[c] | **4a** | $CH_2Cl_2$ | 84 | > 20:1 | 96 |

Reactions were run using **1a** (0.15 mmol), **2** (0.30 mmol), catalyst (0.015 mmol), and solvent (0.30 ml). Yields represent material isolated after silica gel column chromatography. Diastereomeric ratios (dr) were determined by $^1$H NMR spectroscopy
[a]Reaction was run using trimethylsilyl cyanide (0.30 mmol) with 2-propanol (0.30 mmol) instead of **2**
[b]Reaction was run using trimethylsilyl cyanide (0.30 mmol) instead of **2**
[c]Reaction was run using 0.18 mmol of **2** and 0.0075 mmol of **4a** for 48 h
[d]Values are for minor diastereomers

While the cyanohydrin could not be isolated from the reaction using acetone cyanohydrin (**2**) because of the reversibility of ketone cyanation, the reaction using trimethylsilyl cyanide afforded the cyanosilylation product **7** with an enantioselectivity of only 17% *ee* (see also Table 1, entry 9 and Supplementary Fig. 9). These results imply that the enantioselectivity of the formation of **3** was not determined in the nucleophilic 1,2-addition step to form the cyanohydrin, but in a concerted manner via dynamic kinetic resolution involving the asymmetric oxy-Michael addition of cyanohydrins, one enantiomer of which was selectively recognized and activated by the bifunctional organocatalyst.

**Transformation of the product**. The ability of the cyano group to be transformed into various functional groups[36, 37] further increases the value of our products as synthetic intermediates. Reaction of the cyano group enables a range of functional groups to be installed at the tetrasubstituted chiral centre in the pharmaceutically important tetrahydropyrans. The cyano group of **3a** was converted to an aminomethyl group by treatment with lithium aluminium hydride, giving **9** after restoration of the carbonyl moiety without erosion of the optical purity (Fig. 5). Reaction with diisobutylaluminium hydride transformed the cyano group to a formyl group, a useful handle for further

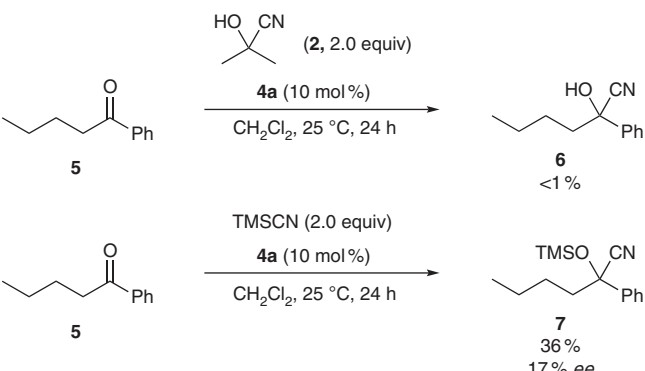

**Fig. 3** Substrate scope. Reactions were run using **1** (0.15 mmol), **2** (0.30 mmol), and **4a** (0.015 mmol) in CH$_2$Cl$_2$ (0.30 ml). Yields represent material isolated after silica gel column chromatography. Diastereomeric ratios (dr) were determined by $^1$H NMR spectroscopy. *Reaction was run for 72 h

transformation, affording **11** in good yield while maintaining the enantiomeric excess.

## Discussion

We demonstrated a concise organocatalytic cycloetherification for the highly enantio- and diastereoselective synthesis of tetrahydropyrans with two chiral centres, one of which is a fully substituted stereogenic carbon centre. This method features an asymmetric intramolecular oxy-Michael addition and dynamic kinetic resolution involving reversible generation of chiral cyanohydrins. The proposed rationale for this transformation entails a chiral bifunctional organocatalyst selectively recognizing a specific chair-like conformation of the intermediate. The weak steric interaction and anomeric effect induced by the cyano group also play important roles in the control of stereoselectivity. The reaction products are useful synthetic intermediates because the

**Fig. 4** Cyanohydrin formation under the optimized conditions. Reactions were run using **5** (0.15 mmol), **2** or trimethylsilylcyanide (0.30 mmol), and **4a** (0.015 mmol) in CH$_2$Cl$_2$ (0.30 ml)

**Fig. 5** Transformations of the cyano group in product **3a**. Synthesis of **8**: **3a** (0.10 mmol) was treated with lithium aluminium hydride (0.50 mmol) in Et$_2$O (1.0 ml). Synthesis of **9**: **8** (0.10 mmol) was treated with manganese dioxide (1.0 mmol) in CH$_2$Cl$_2$ (2.0 ml). Synthesis of **10**: **3a** (0.10 mmol) was treated with diisobutylaluminium hydride (0.40 mmol) in CH$_2$Cl$_2$ (1.0 ml). Synthesis of **11**: **10** (0.056 mmol) was treated with pyridinium chlorochromate (0.17 mmol) in CH$_2$Cl$_2$ (2.0 ml)

cyano group can be further transformed into various functional groups to realize products that have potential as pharmaceutical agents. The current strategy provides an efficient route to a wide range of tetrahydropyran derivatives that are otherwise difficult to access, which will facilitate their evaluation. Further studies regarding the application of this methodology to expand the range of accessible optically active tetrahydropyrans bearing other substitution patterns are currently ongoing in our laboratory and will be reported in due course.

## Methods

**General procedure for the asymmetric synthesis of tetrahydropyrans 3.** To a 5-ml vial were added sequentially α,β-unsaturated substrate **1** (0.15 mmol), CH$_2$Cl$_2$ (0.30 ml), bifunctional catalyst **4a** (0.015 mmol), and acetone cyanohydrin (**2**, 0.30 mmol). The mixture was stirred in an oil bath maintained at 25 °C for 24 h. The reaction mixture was subsequently diluted with hexane/EtOAc (1:1, v/v), passed through a short silica gel pad to remove **4a**, and concentrated in vacuo to give the crude tetrahydropyrans **3**. Purification of the crude products by flash silica gel column chromatography using CH$_2$Cl$_2$/hexane (3:1, v/v) and then hexane/ EtOAc (3:1–10:1, v/v) as an eluent afforded the corresponding tetrahydropyrans **3**.

**Data availability**. Additional data supporting the findings described in this manuscript are available in the Supplementary Information. For full character-ization data of new compounds and experimental details, see Supplementary Methods. For the $^1$H and $^{13}$C NMR spectra of new compounds, see Supplementary Figs. 10–99. For HPLC chromatogram profiles of the reaction products, see Sup-plementary Figs. 100–121. For an ORTEP drawing of **3a**, see Supplementary Fig. 122. X-ray crystallographic data have also been deposited at the Cambridge Crystallographic Data Centre (http://www.ccdc.cam.ac.uk/) with the accession code CCDC 1566029. All other data are available from the authors upon reasonable request.

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

## Acknowledgements
We thank Dr. Hiroyasu Sato (RIGAKU) and Professor Takuya Kurahashi (Kyoto University) for X-ray crystallographic analysis. This work was supported financially by the Japanese Ministry of Education, Culture, Sports, Science and Technology (15H05845 and 16K13994). K.A. also acknowledges the Asahi Glass Foundation, Toyota Physical and Chemical Research Institute, Tokyo Institute of Technology Foundation, the Naito Foundation, Research Institute for Production Development, the Tokyo Biochemical Research Foundation, the Uehara Memorial Foundation, the Kyoto University Foundation, and Kyoto University Research Development Program 2016. A.M. also acknowledges the Japan Society for the Promotion of Science for Young Scientists for fellowship support.

## Author contributions
K.A. conceived and designed the study. S.M. supervised the project. N.Y., Y.F. and A.M. carried out the chemical experiments and analysed the data. K.A. wrote the manuscript. All authors discussed the results and commented on the manuscript.

## Additional information

**Competing interests:** The authors declare no competing financial interests.

