## [Peer Review File · Nature Communications]

Reviewers' comments:

Reviewer #1 (Remarks to the Author):

The paper reported by prof. Seijiro Matsubara and prof. Keisuke Asano described a concise organocatalytic cycloetherification for the highly enantio- and diastereoselective synthesis of tetrahydropyrans, involving simultaneous construction of two chiral centres, one of which is fully substituted. Moderate to good yields, high enantioselectivities, and excellent diastereoselectivities were obtained. More impressively, aliphatic ketones were also successfully transformed. I have some concerns with this paper but think that it could be acceptable for publication in Nature Communications.

Some items the authors should address:

1. No substituents in the double bond of bis-ketone substrates have been tested. The authors need to demonstrate a few examples
2. The applicability of this catalytic system to other classes of substrates, especially for the synthesis of tetrahydrofuran and oxepane, should also be examined.
3. Possibly a weak point in this paper is the mechanism study. It is the highlight (cycloetherification via dynamic kinetic resolution) in this article, however, the authors just conducted some preliminary experiments. The reaction using ketone 5, without any α,β -unsaturated carbonyl moiety afforded <1% cyanosilylation product 6. How about the ee value of 6? The experimental results, from the reaction performed using the racemic compounds which is shown in Figure 2a as an intermediate, will be more direct to prove the author's proposed mechanism.

Overall, this is a good paper and suitable for publication in Nature Communications when the cited items have been answered.

Reviewer #2 (Remarks to the Author):

This paper describes an important cycloetherification for the asymmetric synthesis of chiral tetrahydropyrans. The manuscript is well written with excellent results and the work will be attractive to broad readers, but the following points should be made clear to consider the criteria.

1. On the scope of the reaction, can substrates with aldehydes be used for the asymmetric cyclization?
2. Other cyanation reaction conditions such as previously reported cyanosilylation methods by Feng, Shibasaki, Jacobsen, Ishihara, or Deng?
3. Any theoretical supporting evidence for the proposed dynamic kinetic resolution process?
4. Showing specific usefulness of the methodology for the biologically active compounds will be helpful for understanding the importance of the work.

This paper should be reconsidered after the major modifications of the manuscript. The quality of the results are considered to be high.

Reviewer #3 (Remarks to the Author):

Asano and Matsubara developed a chiral bifunctional organocatalyzed enantioselective approaches to six-membered oxacycles with multiple stereogenic centres. Considering the difficulty in controlling the stereoselectivity in six-membered oxacycles, this strategy provides an efficient approach to a wide range of optically active tetrahydropyrans. However, the author claimed that the catalysts recognize a specific chair-like conformation of the intermediate, in which a cyano-induced anomeric effect also plays a key role in the control of stereoselectivity. I am not convinced

in accepting the above explanation. To me is more of a stereoelectronic effect of aryl/alkyl vs CN group, where aryl/alkyl preferred to stay at equatorial position. Which was exactly observed by them (Chem. Lett. 45, 2016, 1300–1303 and by Gharpure (Eur. J. Org. Chem. 2014, 3570–3574). Although, the results are very good from the synthetic point of view, I am not able to recommend this work for the publication in Nature Communication because of above reason. However, there are few more comments need to be address before submitting to another journal:

- 1) Authors should cite the Eur. J. Org. Chem. 3570-3574 (2014) in this they have prepared cis-2,6-Disubstituted Tetrahydropyrans with diastereoselectivity.
- 2) In figure-2b authors explained matched and mismatched TS, what happening when the same reaction performed with epimeric catalyst.
- 3) Further to confirm anomeric effect authors may have to synthesize 2,5-disubstituted furans under similar reaction conditions.
- 4) Did the authors observe any changes in selectivity in polar solvents (other than tabulated solvents) because they have the ability to reduces the lone pair (anomeric) interactions? Other nucleophiles which does not show anomeric effect (such as alkyl) need to explored.
- 5) In Table 2 all reactions were carried out about 24 hours. Is there any electronic effects of enone moiety on cyclization.
- 6) In figure 4 diastereoselectivity of structure 9 and 11 are missing. Have they maintained the same diastereoselectivity of structures 8 and 10?
- 7) In supporting information, please check HRMS data of compound 3k it has a difference between theoretical and experimental value is around 0.04 and NMR data of 1f and 1i k data showing solvent peaks.

CorrectionTable

Revision Request (1) from Reviewer #1: No substituents in the double bond of bis-ketone substrates have been tested. The authors need to demonstrate a few examples.	Answer: The reactions from substrates bearing trisubstituted olefins were carried out, and Supplementary Figure 3 was added to the SI. However, the reactions failed to yield cyclic products at 25 °C or 50 °C. Further optimization for the use of such substrates is currently underway, and we hope to investigate it as a part of our next work and report it in the near future. Supplementary Figure 3:  Supplementary Figure 3. Reactions of (E)-3-methyl-1,7-diphenylhept-2-ene-1,7-dione ((E)-28) and (Z)-3-methyl-1,7-diphenylhept-2-ene-1,7-dione ((Z)-28). The reactions from substrates (E)- and (Z)-28 bearing trisubstituted olefins were carried out, however, the reactions failed to yield cyclic products at 25 °C or 50 °C. Further optimization for the use of such substrates is currently underway.
--	--

Revision Request
(2) from Reviewer
#1:

The applicability of this catalytic system to other classes of substrates, especially for the synthesis of tetrahydrofuran and oxepane, should also be examined.

Answer:

The synthesis of a tetrahydrofuran (THF) derivative was investigated, and **Supplementary Figure 4** was added to the SI. However, under the present conditions, the stereoselectivities were not as good as those obtained from the reactions affording the tetrahydropyran (THP) derivatives. We believe the anomeric effect is essential for obtaining the excellent diastereoselectivities, but the small steric effect of the cyano group, decreasing 1,3-diaxial interactions in the six-membered oxacycle, also seems to be important. Individual optimization of bifunctional catalysts is also necessary for obtaining higher enantioselectivity in THF synthesis. Thus, we consider the current protocol is useful for the construction of chiral THPs. Further optimization for the synthesis of THFs is currently underway, and we hope to investigate it as a part of our next work and report it in the near future.

Supplementary Figure 4:

Supplementary Figure 4. Reaction of (*E*)-1,6-diphenylhex-2-ene-1,6-dione (**32**).

The synthesis of a tetrahydrofuran (THF) derivative **40** was investigated, however, under the present conditions, the stereoselectivities were not as good as those obtained from the reactions affording the tetrahydropyran (THP) derivatives **3**. We believe the anomeric effect is essential for obtaining the excellent diastereoselectivities, but the small steric effect of the cyano group, decreasing 1,3-diaxial interactions in the six-membered oxacycle, also seems to be important. Individual optimization of bifunctional catalysts is also necessary for obtaining higher enantioselectivity in THF synthesis. Thus, we consider the current protocol is useful for the construction of chiral THPs. Further optimization for the synthesis of THFs is currently underway.

Revision Request
(3) from Reviewer #1:

Possibly a weak point in this paper is the mechanism study. It is the highlight (cycloetherification via dynamic kinetic resolution) in this article, however, the authors just conducted some preliminary experiments. The reaction using ketone **5**, without any α,β -unsaturated carbonyl moiety afforded <1% cyanosilylation product **6**. How about the ee value of **6**? The experimental results, from the reaction performed using the racemic compounds which is shown in Figure 2a as an intermediate, will be more direct to prove the author's proposed mechanism.

Answer:

From the reaction using ketone **5**, "<1%" means that sufficient product was not obtained and thus, the ee of product **6** could not be analysed. Because of this low yield, the corresponding cyanosilylation was carried out, and the ee of product **7** was analysed.

Additionally, we also investigated the reactions from the possible racemic intermediate. Because it is difficult to isolate the cyanohydrin because of the reversibility of its formation, we carried out the reactions using the corresponding racemic silyl ether, and **Supplementary Figure 8** was added to the SI. However, the reaction did not afford the cyclic product and more than 80% of the starting material was recovered. Thus, the reaction mechanism was discussed by referring to the results indicated in Figure 3 of the main manuscript.

Supplementary Figure 8:

Supplementary Figure 8. Reactions from racemic cyanohydrin silyl ether (±)-**39**.

To gain insight into the reaction mechanism, the reaction from the possible racemic intermediate was investigated. Because it is difficult to isolate the cyanohydrin because of the reversibility of its formation, the reaction using the corresponding racemic silyl ether was carried out, however, the reaction did not afford the cyclic product. The reaction did not proceed even in the presence of *i*-PrOH. Thus, the reaction mechanism was discussed by referring to the results indicated in Figure 3 of the main manuscript.

Procedure

To a 5-mL vial were sequentially added (±)-**39** (43.3 mg, 0.115 mmol), CH_2Cl_2 (0.23 mL), and **4a** (5.2 mg, 0.0115 mmol) (in the latter case, additionally 2-propanol (2 equiv) was added). The mixture was stirred in an oil bath maintained at 25 °C for 24 h. The reaction mixture was subsequently diluted with hexane/EtOAc (v/v = 1/1), passed through a short silica gel pad to remove **4a**, and concentrated in vacuo.

Revision Request
(1) from Reviewer
#2:

On the scope of the
reaction, can
substrates with
aldehydes be used
for the asymmetric
cyclization?

Answer:

The reaction from an aldehyde was carried out, and **Supplementary Figure 5** was added to the SI. However, the diastereomeric ratio (dr) was not as good as that of the reactions from ketones under the present conditions. We believe the anomeric effect is essential for obtaining the excellent diastereoselectivities, but the small steric effect of the cyano group compared with another substituent on the same carbon also seems to be important to favor its axial position. Thus, we consider the current protocol is useful for the construction of a tetrasubstituted chiral centre, and supplements the conventional diastereoselective synthesis of less substituted derivatives (such as *Eur. J. Org. Chem.* 3570–3574 (2014)). Further optimization for the use of aldehyde substrates is currently underway, and we hope to investigate it as a part of our next work and report it in the near future.

Supplementary Figure 5:

Supplementary Figure 5. Reaction of (*E*)-7-oxo-7-phenylhept-5-enal (**34**).

The reaction from an aldehyde was carried out, however, the diastereomeric ratio (dr) was not as good as that of the reactions from ketones under the present conditions. We believe the anomeric effect is essential for obtaining the excellent diastereoselectivities, but the small steric effect of the cyano group compared with another substituent on the same carbon also seems to be important to favor its axial position. Thus, we consider the current protocol is useful for the construction of a tetrasubstituted chiral center, and supplements the conventional diastereoselective synthesis of less substituted derivatives.² Further optimization for the use of aldehyde substrates is currently underway.

Revision Request
(2) from Reviewer
#2:

Other cyanation
reaction conditions
such as previously
reported
cyanosilylation
methods by Feng,
Shibasaki,
Jacobsen, Ishihara,
or Deng?

Answer:

The cyanosilylation methods reported by Shibasaki and Jacobsen were applied to our substrate, and **Supplementary Figures 1 and 2** were added to the SI. Under the conditions with $\text{Yb}(\text{O}i\text{-Pr})_3$ as a catalyst corresponding to Shibasaki's protocol (**Supplementary Figure 1**), only the conjugate addition of the cyanide to the enone proceeded in 12% yield. Additionally, under the conditions with $\text{Ti}(\text{O}i\text{-Pr})_4$ as a catalyst (**Supplementary Figure 1**), no reaction took place. Under the conditions with the Jacobsen's chiral thiourea catalyst (**Supplementary Figure 2**), only the 1,2-addition of the cyanide to the ketone proceeded with moderate to good enantioselectivities. Thus, we consider our protocol is essential for the described reactions to proceed.

Supplementary Figure 1:

Supplementary Figure 1. Reactions with $\text{Yb}(\text{O}i\text{-Pr})_3$ or $\text{Ti}(\text{O}i\text{-Pr})_4$ (Shibasaki's protocol³).

The cyanosilylation protocols corresponding to the method reported by Shibasaki group³ were applied to our substrate. Under the conditions with $\text{Yb}(\text{O}i\text{-Pr})_3$ as a catalyst, only the conjugate addition of the cyanide to the enone proceeded in 12% yield. Additionally, under the conditions with $\text{Ti}(\text{O}i\text{-Pr})_4$ as a catalyst, no reaction took place. Thus, we consider our protocol is essential for the described reactions to proceed.

Procedure

To the solution of $\text{Yb}(\text{O}i\text{-Pr})_3$ or $\text{Ti}(\text{O}i\text{-Pr})_4$ (0.015 mmol) in solvent (0.3 mL) was added trimethylsilylcyanide (3.0 μL , 0.03 mmol) in an ice bath, and the mixture was stirred at ambient temperature for 30 min. To the solution were added **1a** (41.8 mg, 0.15 mmol) followed by additional trimethylsilylcyanide (30 μL , 0.3 mmol) at ambient temperature or $-20 \text{ }^\circ\text{C}$. After the mixture was stirred for hours, H_2O (0.2 mL) was added. The reaction mixture was subsequently diluted with hexane/EtOAc ($v/v = 1/1$), passed through a short silica gel pad to remove the catalyst, and concentrated in vacuo. In the case of the reaction using $\text{Yb}(\text{O}i\text{-Pr})_3$, purification of the crude product by flash silica gel column chromatography using hexane/EtOAc ($v/v = 3/1$) was carried out.

Supplementary Figure 2:

entry	T (°C)	yield (%) ^b	ee (%)
1	-78	27	98
2	25	16	60

Reactions were run using **1a** (0.15 mmol), TMSCN (0.33 mmol), CF₃CH₂OH (0.15 mmol), **4h** (0.015 mmol), and CH₂Cl₂ (0.30 mL). Yields represent material isolated after silica gel column chromatography.

Supplementary Figure 2. Reactions with catalyst **4h** (Jacobsen's protocol⁴).

The cyanosilylation protocols corresponding to the method reported by Jacobsen group⁴ were applied to our substrate. Under the conditions with the Jacobsen's chiral thiourea catalyst, only the 1,2-addition of the cyanide affording **39** proceeded with moderate to good enantioselectivities. Thus, we consider our protocol is essential for the described reactions to proceed.

Procedure

To a 5-mL round bottom flask were sequentially added **4h** (2.89 mg, 0.0075 mmol), **1a** (41.8 mg, 0.15 mmol), trimethylsilylcyanide (41 μL, 0.33 mmol) and CH₂Cl₂ (0.3 mL). The reaction mixture was stirred at -78 °C or 25 °C for 15 min. 2,2,2-Trifluoroethanol (11 μL, 0.15 mmol) was then added, and the mixture was stirred at the same temperature for 24 h. After being warmed to ambient temperature, the reaction mixture was subsequently diluted with hexane/EtOAc (v/v = 1/1), passed through a short silica gel pad to remove **4h**, and concentrated in vacuo. Purification of the crude product by flash silica gel column chromatography using hexane/EtOAc (v/v = 5/1) as an eluent afforded **39**.

Supplementary Figure 2 (Continued):

Preparation of racemic 39

To a solution of **1a** (83.6 mg, 0.30 mmol) in CH₃CN (0.3 mL) were added trimethylamine *N*-oxide (4.51 mg, 0.060 mmol) and trimethylsilylcyanide (56 μL, 0.45 mmol) at 25 °C. The solution was stirred for 11.5 h. The reaction mixture was subsequently diluted with hexane/EtOAc (v/v = 1/1), passed through a short silica gel pad to remove trimethylamine *N*-oxide, and concentrated in vacuo. Purification of the crude product by flash silica gel column chromatography using hexane/EtOAc (v/v = 5/1) as an eluent afforded **(±)-39**.

Revision Request (3) from Reviewer #2: Any theoretical supporting evidence for the proposed dynamic kinetic resolution process?	Answer: As mentioned below (Revision Request (4) from Reviewer #3), additional experiments show that polar solvents, which have the ability to reduce lone pair (anomeric) interactions, decrease the diastereoselectivity, and the use of a tertiary alcohol substrate bearing an alkyl group instead of the cyano group resulted in low diastereoselectivity. We believe that these results as well as the results indicated in Figure 3 of the main manuscript indicate there is a kinetic resolution process involving an anomeric effect by the cyano group in the control of stereoselectivity on the tetrasubstituted chiral centre. Additionally, in support of the anomeric effect being involved in providing an axial cyano group, similar discussion was reported in the synthesis of nitrogen-containing six-membered rings (JOC 49, 2392–2400 (1984); JACS 125, 4970–4971 (2003)). These references were added as refs 33 and 34. In light of this correction, the numbering of the following references was also modified. Ref 33: Bonin, M., Romero, J. R., Grierson, D. S. & Husson, H.-P. 2-Cyano-Δ^3-piperideines. 12. Stereochemistry of formation of N-benzyl-2-cyano-Δ^3-piperideines and facile isomerization on alumina to 2-cyano-Δ^4-piperideines. A potentially general route to the synthesis of 2,6-disubstituted piperidine alkaloids. J. Org. Chem. 49, 2392–2400 (1984). Ref 34: Amos, D. T., Renslo, A. R. & Danheiser, R. L. Intramolecular [4 + 2] cycloadditions of iminoacetonitriles: a new class of azadienophiles for hetero Diels–Alder Reactions. J. Am. Chem. Soc. 125, 4970–4971 (2003).
---	--

Revision Request (4) from Reviewer #2: Showing specific usefulness of the methodology for the biologically active compounds will be helpful for understand the importance of the work.	Answer: Page 4, line 6, a sentence was added as follows: “The resulting cyclic structures are found in a variety of bioactive agents^{3–6}; the functionality of the cyano group has not only been utilized for further transformations in the synthesis of such important compounds^{54–59}, but also plays a significant role in their biological activities^{60–62}.” Along with the correction, additional references were added as refs 54–62: (54)Barrero, A. F., Alvarez-Manzaneda Roldán, E. J., Romera Santiago, J. L. & Chahboun, R. Highly diastereoselective synthesis of manoyl oxide derivatives by TiCl₄-catalyzed nucleophilic cleavage of ambracetal derivatives. Synlett 2313–2316 (2003). (55)Tadanier, J., Lee, C.-M., Hengeveld, J., Rosenbrook, W., Jr., Whittern, D. & Wideburg, N. 2-Deoxy-2-(substituted-methyl)analogs of β-Kdop. Carbohydr. Res. 201, 209–222 (1990). (56)Schweizer, F., Otter, A. & Hindsgaul, O. Synthesis of sugar-fused GABA-analogs. Synlett 1743–1746 (2001). (57)Schweizer, F. & Hindsgaul, O. Synthesis of a galacto-configured C-ketoside-based γ-sugar-amino acid and its use in peptide coupling reactions. Carbohydr. Res. 341, 1730–1736 (2006). (58)Pal, A. P. J., Gupta, P., Reddy, Y. S. & Vankar, Y. D. Synthesis of fused oxa-aza spiro sugars from D-glucose-derived δ-lactone as glycosidase inhibitors. Eur. J. Org. Chem. 6957–6966 (2010). (59)Yamada, H., Adachi, M. & Nishikawa, T. Stereocontrolled synthesis of the oxathiabicyclo[3.3.1]nonane core structure of tagetitoxin. Chem. Commun. 49, 11221–11223 (2013). (60)Janero, D. A., Cohen, N., Burghardt, B. & Schaer, B. H. Novel 6-hydroxy chroman-2-carbonitrile inhibitors of membrane peroxidative injury. Biochem. Pharmacol. 40, 551–558 (1990). (61)Ooiwa, H., Janero, D. R., Stanley, A. W. H. & Downey, J. M. Examination of two small-molecule antiperoxidative agents in a rabbit model of posts ischemic myocardial infarction. J. Cardiovasc. Pharmacol. 17, 761–767 (1991). (62)Boscoboinik, D., Özer, N. K., Moser, U., Azzi, A. Tocopherols and 6-hydroxy-chroman-2-carbonitrile derivatives inhibit vascular smooth
--	---

	muscle cell proliferation by a nonantioxidant mechanism. Arch. Biochem. Biophys. 318, 241–246 (1995).
--	--

Revision Request (1) from Reviewer #3: Authors should cite the Eur. J. Org. Chem. 3570-3574 (2014) in this they have prepared cis-2,6-Disubstituted Tetrahydropyrans with diastereoselectivity.	Answer: The suggested reference was added as ref 10. Because of this correction, the numbering of the following references was also modified. Ref 10: Gharpure, S. J., Prasad, J. V. K. & Bera, K. Tandem nucleophilic addition/oxa-Michael reaction for the synthesis of cis-2,6-disubstituted tetrahydropyrans. Eur. J. Org. Chem. 3570–3574 (2014).
Revision Request (2) from Reviewer #3: In figure-2b authors explained matched and mismatched TS, what happening when the same reaction performed with epimeric catalyst.	Answer: With an epimeric catalyst, B is matched, whereas A is mismatched. Actually, 4d/4e and 4f/4g afforded the opposite enantiomers of the same diastereomer (Table 1, entries 4–7). This trend is often observed in the reactions we developed using the same type of catalysts (refs 37–43). The spatial arrangements of the two functional groups in the catalysts, which interact with the intermediates, are important. Although 4d/4e and 4f/4g are not enantiomers but pseudo-enantiomers, the enantiomeric products of the same diastereomer were obtained, respectively.
Revision Request (3) from Reviewer #3: Further to confirm anomeric effect authors may have to synthesize 2,5-disubstituted furans under similar reaction conditions.	Answer: The reaction affording the 2,5-substituted THF was investigated as mentioned above (Revision Request (2) from Reviewer #1), and Supplementary Figure 4 was added to the SI. However, under the present conditions, the dr was not as good as those in the reactions affording the 2,6-substituted THPs. We believe the anomeric effect is essential for obtaining the excellent diastereoselectivities, but the small steric effect of the cyano group, decreasing 1,3-diaxial interactions in a six-membered oxacycle, also seems to be important. Thus, we consider the current protocol is useful for the construction of chiral THPs. Further optimization of the synthesis of THFs is currently underway, and we hope to investigate it as a part of our upcoming work and report it in the near future.

Revision Request (4) from Reviewer #3: Did the authors observe any changes in selectivity in polar solvents (other than tabulated solvents) because they have the ability to reduce the lone pair (anomeric) interactions? Other nucleophiles which does not show anomeric effect (such as alkyl) need to be explored.	Answer: On the basis of the valuable suggestions of the reviewer, several investigations were carried out; entries 16 and 17 were added in Table 1, and Supplementary Figure 7 was added to the SI. We additionally investigated two polar solvents, CH₃CN and EtOH, and they resulted in the decline of diastereoselectivity, while the high enantioselectivities were maintained. To examine an alkyl group instead of the cyano group, the kinetic resolution of a substrate with a tertiary alcohol bearing a methyl group was investigated; the reaction resulted in significantly low diastereoselectivity with low to moderate enantioselectivities. We consider these results support the importance of the anomeric effect by the cyano group in the control of stereoselectivity on the tetrasubstituted chiral centre. Page 4, line 22, a sentence was modified as follows: Solvent optimization studies identified CH₂Cl₂, CHCl₃, and hydrocarbon solvents as affording especially high stereoselectivities with good yields (Table 1, entries 1, 10–12), The reaction could also be carried out using a smaller amount of 2, and lower catalyst loading of 4a, giving the same excellent stereoselectivity, with a slight decrease in the yield (Table 1, entry 16). ↓ Solvent optimization studies identified CH₂Cl₂, CHCl₃, and hydrocarbon solvents as affording especially high stereoselectivities with good yields (Table 1, entries 1, 10–12), while the use of polar solvents, which in general decrease anomeric effects, resulted in decreased yields and diastereoselectivity albeit with high enantioselectivities observed for both diastereomers (Table 1, entries 13–17). The reaction could also be carried out using a smaller amount of 2, and lower catalyst loading of 4a, giving the same excellent stereoselectivity, with a slight decrease in the yield (Table 1, entry 18).
---	---

Table 1:

Table 1. Optimization of reaction conditions.

Entry	Catalyst	Solvent	Yield (%)	dr	ee (%)
1	4a	CH ₂ Cl ₂	99	>20:1	97
2	4b	CH ₂ Cl ₂	95	14:1	-97
3	4c	CH ₂ Cl ₂	<1	—	—
4	4d	CH ₂ Cl ₂	89	14:1	-92
5	4e	CH ₂ Cl ₂	69	17:1	-94
6	4f	CH ₂ Cl ₂	72	11:1	93
7	4g	CH ₂ Cl ₂	82	10:1	94
8*	4a	CH ₂ Cl ₂	84	>20:1	97
9 [†]	4a	CH ₂ Cl ₂	14	>20:1	97
10	4a	CHCl ₃	93	>20:1	97
11	4a	Benzene	93	>20:1	95
12	4a	Toluene	90	>20:1	95
13	4a	Et ₂ O	59	20:1	93
14	4a	THF	15	20:1	96
15	4a	EtOAc	38	17:1	94
16	4a	CH ₃ CN	54	3.6:1	95 (93[§])
17	4a	EtOH	23	9.2:1	96 (88[§])
18[‡]	4a	CH ₂ Cl ₂	84	>20:1	96

Reactions were run using **1** (0.15 mmol), **2** (0.3 mmol), catalyst (0.015 mmol), and solvent (0.3 mL). Yields represent material isolated after silica gel column chromatography. Diastereomeric ratios were determined by ¹H NMR.

*Reaction was run using trimethylsilyl cyanide (0.3 mmol) with 2-propanol (0.3 mmol) instead of **2**. [†]Reaction was run using trimethylsilyl cyanide (0.3 mmol) instead of **2**. [‡]Reaction was run using 0.18 mmol of **2** and 0.0075 mmol of **4a** for 48 h. [§]Values are for minor diastereomers.

Supplementary Figure 7:

Supplementary Figure 7. Kinetic resolution of 37.

In order to examine the anomeric effect by the cyano group, the kinetic resolution of a tertiary alcohol **37** bearing a methyl group, which does not show anomeric effects, was investigated; the reaction resulted in significantly low diastereoselectivity with low to moderate enantioselectivities. We consider these results support the importance of the anomeric effect by the cyano group in the control of stereoselectivity on the tetrasubstituted chiral center.

Additionally, in support of the anomeric effect being involved in providing an axial cyano group, similar discussion was reported in the synthesis of nitrogen-containing six-membered rings (*JOC* **49**, 2392–2400 (1984); *JACS* **125**, 4970–4971 (2003)) as mentioned above (Revision Request (3) from Reviewer #2). These references were added as refs 33 and 34.

Revision Request
(5) from Reviewer
#3:

In Table 2 all reactions were carried out about 24 hours. Is there any electronic effects of enone moiety on cyclization.

Answer:

All reactions shown in Table 2 of the main manuscript were carried out for 24 h. We also investigated the reactions of **1a** and **1b** for a shorter time, 6 h, and **Supplementary Figure 6** was added to the SI. The results indicated that there are electronic effects of the enone moieties on the reaction rate (86% yield and 22% yield, respectively).

Supplementary Figure 6:

entry	R	yield (%)	dr	ee (%)
1	Ph (1a)	86	>20:1	97
2	4- $\text{CH}_3\text{OC}_6\text{H}_4$ (1b)	22	>20:1	93

Reactions were run using **1** (0.15 mmol), **2** (0.3 mmol), **4a** (0.015 mmol), and CH_2Cl_2 (0.3 mL). Yields represent material isolated after silica gel column chromatography.

Supplementary Figure 6. Investigations of electronic effects of enone moieties.

All reactions shown in Table 2 of the main manuscript were carried out for 24 h. The reactions of **1a** and **1b** were also investigated for a shorter time, 6 h, and the results indicated that there are electronic effects of the enone moieties on the reaction rate.

Revision Request
(6) from Reviewer #3:

In figure 4 diastereoselectivity of structure 9 and 11 are missing. Have they maintained the same diastereoselectivity of structures 8 and 10?

Answer:

Yes, **9** and **10** were obtained as single diastereomers, although **8** and **10** were diastereomer mixtures with respect to the stereochemistry of the secondary alcohol. For the sake of clarity, Figure 4 was corrected as follows.

Revision Request
(7) from Reviewer #3:

In supporting information, please check HRMS data of compound 3k it has a difference between theoretical and experimental value is around 0.04 and NMR data of 1f and 1i k data showing solvent peaks.

Answer:

HRMS data of compound **3k**:

The value was wrong. We corrected it as follows.

“HRMS Calcd for $\text{C}_{18}\text{H}_{23}\text{NO}_2\text{Na}$: $[\text{M}+\text{Na}]^+$, 308.1621. Found: m/z 308.1149.”

↓

“HRMS Calcd for $\text{C}_{18}\text{H}_{23}\text{NO}_2\text{Na}$: $[\text{M}+\text{Na}]^+$, 308.1621. Found: m/z 308.1616.”

NMR data of **1f**, **1i**, and **1k** data:

For **1f** and **1i**, the solvents were further removed *in vacuo*, the NMR spectra were obtained again and added to the supporting information. We believe the NMR of **1k** is now suitable, but if there are any other problems, we are willing to further investigate them.

Additional Revision (1):	During the revision process, another colleague participated in the additional experiments. Thus, we would like to add an additional author. Author names (the main manuscript and the SI): The notes were modified as follows: “Naoki Yoneda, Yuki Fujii, Keisuke Asano* and Seiji Matsubara*” ↓ “Naoki Yoneda, Yuki Fujii, Akira Matsumoto, Keisuke Asano* and Seiji Matsubara*” Author contributions: “N.Y. and Y.F. carried out the chemical experiments and analysed the data.” ↓ “N.Y., Y.F., and A.M. carried out the chemical experiments and analysed the data.”
Additional Revision (2):	Acknowledgements: “K.A. also acknowledges the Asahi Glass Foundation, Toyota Physical and Chemical Research Institute, Tokyo Institute of Technology Foundation, and the Naito Foundation.” ↓ “K.A. also acknowledges the Asahi Glass Foundation, Toyota Physical and Chemical Research Institute, Tokyo Institute of Technology Foundation, the Naito Foundation, Research Institute for Production Development, the Tokyo Biochemical Research Foundation, the Uehara Memorial Foundation, and the Kyoto University Foundation. A.M. also acknowledges the Japan Society for the Promotion of Science for Young Scientists for the fellowship support.”
Additional Revision (3):	Results: Subheadings were added as follows: “Optimization of reaction conditions.” “Substrate scope.” “Mechanistic insights.” “Transformation of the product.”

Additional Revision (4):	SI, page S164: Supplementary References were added as follows: Supplementary References  1. Vakulya, B., Varga, S., Csámpai, A. & Soós, T. Highly enantioselective conjugate addition of nitromethane to chalcones using bifunctional cinchona organocatalysts. Org. Lett. 7, 1967–1969 (2005). 2. Gharpure, S. J., Prasad, J. V. K. & Bera, K. Tandem nucleophilic addition/oxa-Michael reaction for the synthesis of cis-2,6-disubstituted tetrahydropyrans. Eur. J. Org. Chem. 3570–3574 (2014). 3. Hamashima, Y., Kanai, M. & Shibasaki, M. Catalytic enantioselective cyanosilylation of ketones. J. Am. Chem. Soc. 122, 7412–7413 (2000). 4. Fuerst, D. E. & Jacobsen, E. N. Thiourea-catalyzed enantioselective cyanosilylation of ketones. J. Am. Chem. Soc. 127, 8964–8965 (2005). 																														
Additional Revision (5):	All NMR spectra, HPLC chromatogram profiles, and ORTEP drawing are labelled with “Supplementary Figure XX”, and each Supplementary Figure is cited in the main text.																														
Additional Revision (6):	The “Contents” of the SI (the first page) were corrected as follow:   Contents   Instrumentation and Chemicals S2   Experimental Procedure S3   Characterization Data of Products S19   NMR Spectra (¹H, ¹³C) of Products S30   HPLC Chromatogram Profiles S96   ORTEP Drawing of 3a S113   ↓   Contents   Instrumentation and Chemicals S2   Experimental Procedure S3   Characterization Data of Products S35   NMR Spectra (¹H, ¹³C) of Products S49   HPLC Chromatogram Profiles S139   ORTEP Drawing of 3a S161  	Contents		Instrumentation and Chemicals	S2	Experimental Procedure	S3	Characterization Data of Products	S19	NMR Spectra (¹H, ¹³C) of Products	S30	HPLC Chromatogram Profiles	S96	ORTEP Drawing of 3a	S113	↓		Contents		Instrumentation and Chemicals	S2	Experimental Procedure	S3	Characterization Data of Products	S35	NMR Spectra (¹H, ¹³C) of Products	S49	HPLC Chromatogram Profiles	S139	ORTEP Drawing of 3a	S161
Contents																															
Instrumentation and Chemicals	S2																														
Experimental Procedure	S3																														
Characterization Data of Products	S19																														
NMR Spectra (¹H, ¹³C) of Products	S30																														
HPLC Chromatogram Profiles	S96																														
ORTEP Drawing of 3a	S113																														
↓																															
Contents																															
Instrumentation and Chemicals	S2																														
Experimental Procedure	S3																														
Characterization Data of Products	S35																														
NMR Spectra (¹H, ¹³C) of Products	S49																														
HPLC Chromatogram Profiles	S139																														
ORTEP Drawing of 3a	S161																														

Additional Revision (7):	Chemical structures in the SI were drawn using a Nature Chemistry Chemdraw template.
Additional Revision (8):	SI, page S2: Section title " Supplementary Method " was added.
Additional Revision (9):	Section titles were corrected as follows: "Results and discussion" → "Results" "Conclusion" → "Discussion"
Additional Revision (10):	Methods: Section " Data availability " was added as follows: " Data availability. Additional data supporting the findings described in this manuscript are available in the Supplementary Information. For full characterization data of new compounds and experimental details, see Supplementary Methods. For the ¹ H and ¹³ C NMR spectra of new compounds, see Supplementary Figs 9–98. For HPLC chromatogram profiles of the reaction products, see Supplementary Figs 99–120. For an ORTEP drawing of 3a , see Supplementary Fig. 121."
Additional Revision (11):	Methods: The subheading was corrected as follows: " General procedure for the asymmetric synthesis of tetrahydropyrans 3 To a 5-mL ..." ↓ " General procedure for the asymmetric synthesis of tetrahydropyrans 3. To a 5-mL ..."

Figure 2:

Figure 2 was corrected as follows:

Additional Revision

(13):

Some English words were modified as follows.

Page 1, line 9:

“tetrahydropyrans, involving”

↓

“tetrahydropyrans involving”

Page 1, line 13:

“control of stereoselectivity”

↓

“control of the stereoselectivity”

“intermediates, as”

↓

“intermediates as”

Page 1, line 17:

“cyclic compounds has been”

↓

“cyclic compounds is”

Page 1, line 21:

“In addition”

↓

“Additionally”

“the absolute configurations of such frameworks is”

↓

“the absolute configurations of these frameworks are”

Page 2, line 7:

“which are unable to be”

↓

“which cannot be”

Page 2, line 8:

“conditions, not suitable”

↓

“conditions that are not suitable”

Page 3, line 4:

“carbon nucleophile, as”

↓

“carbon nucleophile as”

Page 3, line 6:

“due to”

↓

“because of”

Page 3, line 8:

“accompanied with”

↓

“accompanied by”

Page 4, line 4:

“This immediately”

↓

“This complex immediately”

Page 4, line 16:

“bifunctionality”

↓

“the bifunctionality”

Page 8, line 2:

“As was expected”

↓

“As expected”

Page 8, line 4:

“In addition”

“Additionally”

Page 8, line 5:

“the chiral bifunctional catalyst”

“a chiral bifunctional catalyst”

Page 8, line 7:

“due to”

“because of”

Page 8, line 11:

“asymmetric oxy-Michael addition”

“the asymmetric oxy-Michael addition”

Page 9, line 10:

“a highly enantio- and diastereoselective synthesis”

“the highly enantio- and diastereoselective synthesis”

Page 9, line 18:

“difficult-to-access”

“difficult to access”

Page 9, line 19:

“derivatives, to”

“derivatives to”

REVIEWERS' COMMENTS:

Reviewer #1 (Remarks to the Author):

In their revised manuscript, Dr Asano and co-workers have taken on board the comments and criticism of this reviewer (and others) and have produced an improved report of their interesting work, fully suitable for publication as an Article in Nature Communications . Thus, this reviewer commends them for being receptive to constructive input and duly supports the publication of their work in Nature Communications.

Reviewer #2 (Remarks to the Author):

I examined the revised manuscript and the correction table by the authors carefully and I found that the revision requests by the reviewers are well answered and the revision is considered to fulfill the requirements for the criteria of acceptance. I would like to recommend this revised manuscript to be published in the journal essentially as it stand.

Reviewer #3 (Remarks to the Author):

Considering the difficulty in controlling the stereoselectivity in six-membered oxacycles, this strategy provides an efficient approach to a wide range of optically active tetrahydropyrans. The results are highly important. The authors have addressed many comments. But, my major concern where the authors claimed that the catalysts recognizes a specific chair-like conformation of the intermediate, in which a cyano-induced anomeric effect also plays a key role in the control of stereoselectivity is not addressed. To me is more of a stereoelectronic effect of aryl/alkyl vs CN group, where aryl/alkyl preferred to stay at equatorial position. The energy difference between axial and equatorial conformers ($-\Delta G_0$): Ph (12 KJ/mol) and CN (0.8 KJ/mol) in substituted cyclohexane. The difference ($-\Delta\Delta G_0$) is 11.2 KJ/mol which is more than sufficient to provide an exclusive formation of equatorial Ph and forcing CN to be axial. Moreover, another methyl keto substitution situated at cis to Ph group. Therefore, the analysis of 'cyano-induced anomeric effect plays a key role in the control of stereoselectivity' is not convincing to this reviewer.

Response to Referees

Revision Request (1) from Reviewer #3: my major concern where the authors claimed that the catalysts recognizes a specific chair-like conformation of the intermediate, in which a cyano-induced anomeric effect also plays a key role in the control of stereoselectivity is not addressed. To me is more of a stereoelectronic effect of aryl/alkyl vs CN group, where aryl/alkyl preferred to stay at equatorial position. The energy difference between axial and equatorial conformers ($-\Delta G^0$): Ph (12 KJ/mol) and CN (0.8 KJ/mol) in substituted cyclohexane. The difference ($-\Delta\Delta G^0$) is 11.2 KJ/mol which is more than	Response: We should agree with the valuable suggestion from reviewer #3 considering the conformational energies ($-\Delta G^0$, A values) known for monosubstituted cyclohexanes. At the same time, although the A values of Me and CN groups are 7.28 and 0.84 kJ/mol, respectively (see ref. 32), which corresponds to an equatorial/axial ratio of ca. 13:1, the formation of 3n resulted in 19:1 dr in our reaction (see Figure 3); therefore, the anomeric effect also contributes to the selectivity. Thus, the manuscript was modified to state that the stereoselectivities are controlled by the small steric interaction and anomeric effect induced by the CN group as well as the chiral bifunctional organocatalyst selectively recognizing a specific chair-like conformation of the intermediate. The following changes were made to the manuscript: Title: “Organocatalytic enantio- and diastereoselective cycloetherification via dynamic kinetic resolution aided by anomeric effects” was changed to ↓ “Organocatalytic enantio- and diastereoselective cycloetherification via dynamic kinetic resolution of chiral cyanohydrins” Page 1, line 17: “in which a cyano-induced anomeric effect also plays a key role in the control of the stereoselectivity” was changed to ↓ “in which the small steric effect of the linear cyano group as well as its anomeric effect play important roles in controlling stereoselectivity”
--	--

sufficient to provide an exclusive formation of equatorial Ph and forcing CN to be axial. Moreover, another methyl keto substitution situated at cis to Ph group. Therefore, the analysis of 'cyano-induced anomeric effect plays a key role in the control of stereoselectivity' is not convincing to this reviewer.

Figure 1:

was changed to ↓

Page 2, line 55:

“we aimed to exploit the anomeric effect¹⁴. This favours an electronegative substituent adjacent to the heteroatom in an axial position in a six-membered oxacycle, enabling efficient orbital interaction with the oxygen atom.”

was changed to ↓

“we aimed to use a small electronegative carbon nucleophile. These features favour the introduction of the substituent adjacent to the heteroatom in an axial position in a six-membered oxacycle, enabling weaker 1,3-diaxial interactions as well as a favourable orbital interaction with the oxygen atom (anomeric effect)¹⁴.”

Figure 2:

was changed to ↓

Page 3, line 67:

“the cyano group is capable of inducing an anomeric effect because of its electronegativity”

was changed to ↓

“the cyano group is known to have a small A value (conformational energy)³² and is capable of inducing an anomeric effect because of its electronegativity”

Along with the above corrections, an additional reference was added as ref 32. Because of this correction, the numbering of the subsequent references was also modified.

Ref 32:

Stereochemistry of Organic Compounds (eds Eliel, E. L. & Wilen, S. H.) Ch. 11 (John Wiley & Sons, 1994).

Page 4, line 76:

“also favoured by the anomeric effect”

was changed to ↓

“also favoured by the weak 1,3-diaxial interactions and anomeric effect”

Page 6, line 131:

“despite the small difference in size between a methyl group and a cyano group, reinforcing the strong influence of the anomeric effect”

was changed to ↓

“despite the relatively small difference in size between methyl and cyano groups, reinforcing that assistance was provided by the anomeric effect”

Page 9, line 182:

“The anomeric effect induced by the cyano group also plays an important role”

was changed to ↓

“The weak steric interaction and anomeric effect induced by the cyano group also play important roles”

Supplementary Figure 4:

“We believe the anomeric effect is essential for obtaining the excellent diastereoselectivities, but the small steric effect of the cyano group, decreasing 1,3-diaxial interactions in the six-membered oxacycle, also seems to be important.”

was changed to ↓

“Although we believe the anomeric effect also assists with obtaining the excellent diastereoselectivities, the small steric effect of the cyano group, which weakens 1,3-diaxial interactions in the six-membered oxacycle, seems to be essential.”

Supplementary Figure 5:

“We believe the anomeric effect is essential for obtaining the excellent diastereoselectivities, but the small steric effect of the cyano group compared with another substituent on the same carbon also seems to be important to favor its axial position.”

was changed to ↓

“Although we believe the anomeric effect also assists with obtaining the excellent diastereoselectivities, the small steric effect of the cyano group compared with that of another substituent on the same carbon seems to be essential to favour its axial position.”